# GeVLM: 3D Object Grounding with Geometry-Enhanced Vision Language Model

## Abstract

Understanding 3D scenes with point cloud data in tasks such as object referencing, question-answering, and captioning poses significant challenges to vision language models (VLMs), due to the complexity of integrating both linguistic and spatial information. While existing methods have mapped point cloud features into LLM space to enable 3D scene comprehension, they often overlook viewpoint information and the relative spatial distance between objects, this can lead to confusion in interpreting spatial descriptions and grounding objects. This paper presents a geometry-enhanced vision LM (GeVLM) to address these challenges. Specifically, we propose viewpoint-consistent position encoding (VCPE) and distance-aware cross-entropy (DACE) loss, which enhance the model's ability to interpret relative spatial relationships agnostic to camera viewpoint and incorporate distance information in the label space. We additionally introduce the DetailedScanRefer dataset, which provides identifiers and spatial annotation for each object mentioned in the referencing description to further emphasize spatial relationships. GeVLM demonstrates significant improvements over the Chat-3D v2 baseline, particularly with 4.0% and 2.7% absolute increase in Acc@0.25 and Acc@0.50 respectively on the ScanRefer benchmark.

## 1 Introduction

The rapid advancement of Multimodal Large Language Models (LLMs) has greatly enhanced their capabilities in addressing a wide range of tasks involving complex input modalities, such as audio (Tang et al., 2024a; Chu et al., 2023; Gong et al., 2024), images (Liu et al., 2024c;b; Li et al., 2023; Bai et al., 2023; Lin et al., 2023; Chen et al., 2023b) and videos (Zhang et al., 2023a; Cheng et al., 2024; Sun et al., 2024). Recent studies have focused on extending the application of LLMs to the understanding of realistic 3D scenes represented by point clouds(Han et al., 2023; Hong et al., 2023; Wang et al., 2023b; Huang et al., 2023b; Chen et al., 2024b;a), enabling these models to perform tasks such as question-answering, object referencing, and captioning for real-world 3D scenes. Specifically, the task of 3D referencing (Chen et al., 2020) requires LLMs to comprehend detailed object descriptions while simultaneously understanding complex 3D scenes to accurately identify the object being referenced. This task presents significant challenges due to the need for a comprehensive understanding of both linguistic and spatial information.

Previous work in this area has successfully grounded LLMs on 3D point clouds, demonstrating scene comprehension abilities (Hong et al., 2023; Han et al., 2023). Chat-3D (Wang et al., 2023b) maps 3D features into the LLM space and uses a relation module to capture spatial relationships, showcasing strong conversational abilities within 3D environments. Further advancements (Huang et al., 2023b) enhance 3D object referencing by integrating unique identifiers with detailed scene annotations. Nevertheless, these approaches overlook the importance of viewpoint consistency across different examples by using 3D world coordinates as input. Moreover, the training objectives are simply cross-entropy (CE) which penalizes other objects equally regardless of whether they are close or far to the target. These along with suboptimal positional encodings limit performance in scene understanding and object grounding.

To address the aforementioned deficiencies, this paper proposes a geometry-enhanced visual language model (GeVLM) to improve 3D object grounding performance from perspectives including model structure, training criteria and dataset annotations. Specifically, we propose the viewpoint-

consistent position encoding (VCPE) which allows relative spatial relationships, e.g. left/right, to be correctly referred to under arbitrary camera viewpoint. Besides, we propose distance-aware cross-entropy (DACE) loss which incorporates relative distance information into the label space so that non-target object tokens receive different levels of penalization depending on their spatial affinity to the target. To further boost the 3D grounding performance, we propose the DetailedScanRefer dataset which includes the object identifier and the location for each object mentioned in the description. As a result, GeVLM showed consistent improvements over the Chat-3D-v2 on a range of 3D scene understanding tasks. Specifically, GeVLM achieved 4.0% and 2.7% absolute improvements in Acc@0.25 and Acc@0.50 on ScanRefer respectively.

The main contributions of this paper are summarized as follows.

- This paper presents GeVLM, a geometry-enhanced VLM for 3D object referencing and understanding, leveraging easy-to-refer object identifiers. To the best of our knowledge, GeVLM is the first visual LLM that formally investigates and incorporates 3D viewpoint information and relative 3D spatial distance in visual LLMs.

- We propose VCPE to ensure position encoding consistency of point cloud coordinates across different viewpoints. In addition, we propose DACE to inject distance information into label space for improved grounding. We also curate the DetailedScanRefer dataset with fine-grained identifier annotations for each object in the description.

- GeVLM demonstrated consistent improvements over the Chat-3D-v2 baseline across various 3D scene understanding tasks, particularly achieving absolute improvements of 4.0% in Acc@0.25 and 2.7% in Acc@0.50 on the ScanRefer benchmark.

## 2 RELATED WORK

**3D Grounding using Language Models**  Recent research has explored the integration of Large Language Models (LLMs) with 3D object understanding for various applications. LLM-Grounder (Yang et al., 2024) utilizes LLMs to decompose complex queries and evaluate spatial relations for zero-shot 3D visual grounding. Grounded 3D-LLM (Chen et al., 2024b) introduces scene referent tokens and contrastive language-scene pre-training to unify various 3D vision tasks within a generative framework. Transcrib3D (Fang et al., 2024) brings together 3D detection methods and the emergent reasoning capabilities of large language models (LLMs). Cube-LLM (Cho et al., 2024), a multi-modal large language model, can ground and reason about 3D objects in images without 3D-specific architectural design or training.

**Language-Driven 3D Scene Understanding**  There has been growing interest in using natural language to enhance how computers interpret and interact with 3D environments. This approach, known as "3D-language scene understanding," involves training models to understand 3D scenes based on verbal instructions. This method is applied across several tasks. Specifically, 3D Visual Grounding (Chen et al., 2020; Huang et al., 2022; Wang et al., 2023a; Hsu et al., 2023; Yang et al., 2024; Unal et al., 2024) involves models identifying a specified object within a 3D scene according to a language query.

**3D Large Multi-modal Models**  Through the usage of large scale 3D object datasets (Yu et al., 2022; Xue et al., 2023; Zhou et al., 2023), 3D Object-level Large Multi-modal Models (LMMs)(Xu et al., 2023; Liu et al., 2024a; Qi et al., 2024; Tang et al., 2024b) have managed to bridge the gap between 3D modality and texts. However, these models fall short when complex spatial reasoning is needed for 3D scenes. Therefore, multiple models (Ziyu et al., 2023; Wang et al., 2023b; Huang et al., 2023b; Chen et al., 2024b) have been proposed as scene-level LLMs. 3D-LLM (Hong et al., 2023) uses point clouds and their features as input and can handle various 3D-related tasks. The model attempts to improve the understanding of complex spatial relationships among objects by using positional embeddings and learning location tokens. However, the model projects 3D features into the 2D feature space of a pretrained vision-language model, posing significant challenges to capture the 3D spatial structure and complex relationships among objects. Chat-3D (Wang et al., 2023b) and Chat-Scene (Huang et al., 2023a) directly utilizes 3D scene-text data to align the 3D scene with large language model (Llama). However, Chat-3D could only handle one target object per conversation. To overcome this limitation, Chat-3D-v2 (Huang et al., 2023b), as our baseline model,

introduced unique object identifiers in addition to 3D object features, and significantly improved the 3D grounding performance.

# 3 GEOMETRY-ENHANCED VISUAL LM

In this section, we introduce a novel geometry-enhanced vision-language model (GeVLM) designed to address 3D object grounding tasks. Our approach builds on the principles of Chat-3D, leveraging object identifiers for efficient reference within a 3D point cloud. GeVLM integrates a 3D segmenter, a 3D feature encoder, a 3D position encoder, and a pretrained language model (LLM). Notably, the 3D segmenter and feature encoder remain frozen during training. The primary goal is to fine-tune the pretrained LLM to interpret language referring expressions by incorporating 3D geometric cues.

These geometric cues are considered in two key aspects. First, we propose a viewpoint-consistent position encoding (VCPE) to account for camera perspective in 3D scene understanding, as detailed in Section 3.1. Second, we introduce a distance-aware cross-entropy (DACE) loss, discussed in Section 3.2, to highlight the importance of spatial affinity in the grounding task. Additionally, in Section 4, we present a densely annotated grounding dataset, curated with assistance from GPT-4o.

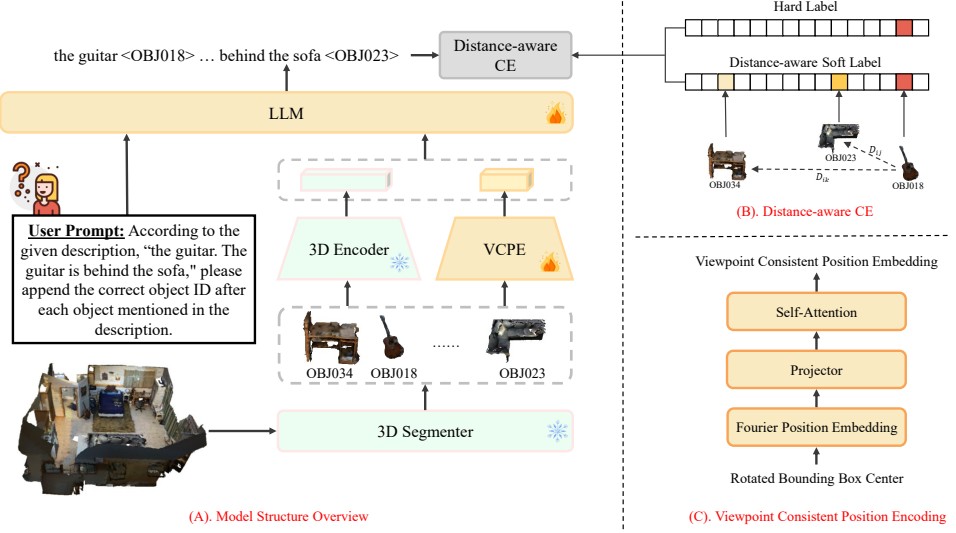

Figure 1: The model structure of the GeVLM (A) together with distance-aware CE loss (B) and an illustration of the viewpoint consistent position encoding (C).

## 3.1 VIEWPOINT-CONSISTENT POSITION ENCODING

Imagine you are inside a room and refer to the chair in front of you. The success of the referring depends on the viewpoint of the observer. In other words, ambiguities will arise if the viewpoint is not known. This is supported by the fact that the annotation from ScanRefer includes the camera pose information. However, existing methods (Wang et al., 2023b; Huang et al., 2023b) overlook the viewpoint information, hence refer to the same 3D point cloud when querying different referring descriptions. We notice that the incorrect grounding outcomes are mainly due to the rotation of the camera viewpoint, which makes relative spatial descriptions such as left/right and front/back confusing to LLMs. For example, in the 3D scenein Fig.2 with 4 different viewpoints, the description "the shelf is to the right of the bed" only makes sense when observing the scene from a consistent viewpoint, e.g. 1 and 4. Nevertheless, methods like Chat-3D and 3D-LLM ignore camera viewpoint, and directly utilize world coordinates as input for object grounding. This inevitably introduces viewpoint inconsistency to the model training and leads to sub-optimal performance.

In GeVLM, we carefully transform 3D point cloud to ensure viewpoint consistency across referring expressions. Based on the transformed coordinates, we propose a position encoding module, VCPE, to effectively learn the relative spatial relationship for downstream 3D tasks. Specifically, to achieve viewpoint consistency, we apply a 3D transformation using the rotation matrix $\mathbf{R} \in \mathbb{R}^{3 \times 3}$ from

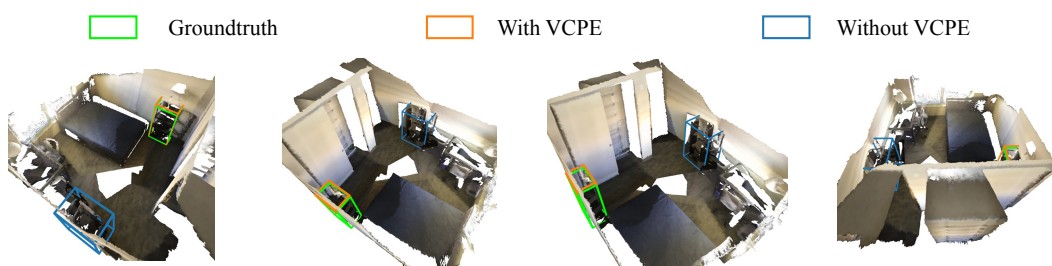

*Description:* It's a black three levels shelf. It is located to the **right** of the bed

Figure 2: Example scene where viewpoint consistency is important. The target shelf in the description is only **right** to the bed in the first viewpoint, and the description confuses the model when using other viewpoints, resulting in an incorrect grounding outcome.

the camera's extrinsic parameters. The translation vector is omitted to maintain a consistent scene scale across different datasets and tasks. For an object, its centre point $\mathbf{v} \in \mathbb{R}^3$ is transformed to $\mathbf{v}_{\text{rot}} = \mathbf{R}\mathbf{v}$. This transformation preserves the spatial configuration of objects relative to the camera orientation and aligns with the viewpoint-dependent language description. As a result, VCPE is crucial for VLMs to effectively generalize across varying viewpoints.

To capture complex spatial relationships, we apply Fourier Feature Mapping (Tancik et al., 2020) to map the low-dimensional coordinates $\mathbf{v}_{\text{rot}}$ to capture high-frequency details as shown in Eqn (1):

$$\gamma(\mathbf{v}_{\text{rot}}) = [\sin(2\pi\mathbf{B}\mathbf{v}_{\text{rot}}), \cos(2\pi\mathbf{B}\mathbf{v}_{\text{rot}})], \tag{1}$$

where $\mathbf{B} \in \mathbb{R}^{3 \times D}$ is a Gaussian random matrix, and $D$ is the dimensionality of the Fourier features. This mapping projects the rotated points into a higher-dimensional space, enabling the model to represent positional information with high-frequency components. The Fourier embeddings for all objects are concatenated into a matrix $\mathbf{F}^{\text{F}} \in \mathbb{R}^{O \times 2D}$, where $O$ represents the number of objects. These embeddings are then projected throught a linear layer, followed by a Gaussian Error Linear Unit (GELU) activation, as shown in Eqn (2):

$$\mathbf{F}^{\text{proj}} = \text{GELU}(\mathbf{F}^{\text{F}}\mathbf{W} + \mathbf{b}), \tag{2}$$

where $\mathbf{W} \in \mathbb{R}^{2D \times D'}$ and $\mathbf{b} \in \mathbb{R}^{D'}$ are learnable parameters, and $D'$ is the dimensionality of the projected features. To capture relative positional dependencies between objects, we further process the projected embeddings using a multi-head self-attention layer:

$$\mathbf{F}^{\text{attn}} = \text{MHSA}(\mathbf{F}^{\text{proj}}) \tag{3}$$

where $\text{MHSA}(\cdot)$ denotes multi-head self-attention. This produces attention-weighted embeddings $\mathbf{F}^{\text{attn}} \in \mathbb{R}^{O \times D'}$ that effectively capture spatial relationships. By integrating these components, VCPE improves the model's capacity to comprehend complex spatial configurations in 3D scenes.

## 3.2 DISTANCE-AWARE CROSS-ENTROPY LOSS

Most 3D VLMs commonly rely on language loss to fine-tune 3D tasks due to its simplicity. Efforts have been made to unify multimodal tasks under a single language-based objective. However, we argue that applying standard cross-entropy (CE) loss to 3D grounding tasks is inadequate. Specifically, when training a model with CE to predict the token for a referred object, it penalizes all other object tokens equally. This contrasts with 3D object detection and segmentation, where the goal is to minimize the distance between the ground truth and predictions.

Building on this insight, we propose DACE to incorporate geometric distance between objects into the loss computation. This approach allows spatial relationships to be considered during training. We categorize tokens into *regular tokens* and *object tokens*. We append 100 object tokens to represent scene objects with token IDs ranging from 32,000 to 32,099. For instance, object token <OBJ000> will be indexed by 32,000. The DACE loss differentiates regular tokens and object token: standard CE loss is applied to the regular tokens, while a soft label is used for the object token predictions. We further detail the DACE loss next.

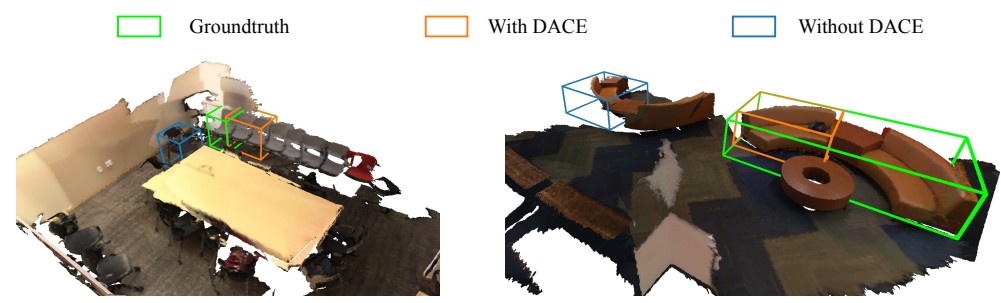

Description: The chair is the first one on the left out of a row of chairs. It has four legs and a bent seat.

Description: The half circle couch is brown and has a wood separator. The couch sits behind a circular coffee table.

Figure 3: Example showing the impact of applying DACE. Left: With DACE, GeVLM correctly focuses on the left end of the row of chairs rather than a random chair. Right: With DACE, GeVLM accurately targets the sofas near the coffee table, avoiding confusion with other similar sofas.

As shown in part (B) of Fig.1, for each scene, we precompute a distance matrix $\mathbf{D} \in \mathbb{R}^{\mathcal{V}_{\text{obj}} \times \mathcal{V}_{\text{obj}}}$ where $D_{ij}$ denotes the Euclidean distance between objects $i$ and $j$, and $\mathcal{V}_{obj}$ represents the number of object tokens. Then, the DACE loss is defined in Eqn. (4).

$$\mathcal{L}_{\text{dist}} = \frac{1}{L} \sum_{i=1}^{L} \boldsymbol{m}_i \cdot \text{CE}(\mathbf{w}_i, P_\theta(\mathbf{y}_i|X_i)) + (1 - \boldsymbol{m}_i) \cdot \text{CE}(\hat{\mathbf{y}}_i, P_\theta(\mathbf{y}_i|X_i)), \tag{4}$$

where $\boldsymbol{m}_i = 1$ for object tokens and $\boldsymbol{m}_i = 0$ for regular tokens. $\text{CE}(\cdot)$ denotes the cross-entropy loss, $L$ is the total number of tokens in the sequence, and $\hat{\mathbf{y}}_i$ is the one-hot label vector. The distance-aware soft label vector, $\boldsymbol{w}_i$, is computed as shown in Eqn. (5).

$$\boldsymbol{w}_i = \exp(-D_{ij}/T) / \sum_{k=1}^{\mathcal{V}_{obj}} \exp(-D_{ik}/T) \tag{5}$$

where $D_{ij} \in [0, 1]$ is the min-max normalized distance between object $i$ and object $j$, and $T$ is the temperature parameter controlling the sharpness of the soft label. The intuition behind this loss is to encourage the model to focus on objects with close affinity, rather than on more distant yet semantically similar objects. This is particularly useful in scenarios with multiple similar objects, such as chairs in a meeting room, where a specific chair is being referred to, as shown in Fig.3.

## 4 DETAILEDSCANREFER: A DENSELY ANNOTATED GROUNDING DATASET

Chat-3D-v2 (Huang et al., 2023b) have utilized object IDs (e.g. <OBJ013>) to refer to specific objects. Such models have shown enhanced spatial reasoning capability. However, only the target objects are labelled with the corresponding identifiers. To further improve grounding, we introduce the DetailedScanRefer dataset, an extension of the ScanRefer dataset (Chen et al., 2020). Detailed-ScanRefer features annotations for both target and landmark objects. Unique object identifiers (object IDs) are assigned to each object in the 3D scene. In DetailedScanRefer, we match all mentioned objects in the description to object IDs from Mask3D (Schult et al., 2023) to maintain consistency and clarity in object references. The dataset generation pipeline is shown in Fig.4.

**Scene Image Retrieval via Camera Pose Matching** Since the quality of images directly rendered from the 3D point cloud is poor, we opted to retrieve a photo of the real-world scene with the most similar view from the ScanNet dataset (Dai et al., 2017)[1]. For each description in ScanRefer, we retrieve the closest camera pose from ScanNet, along with its corresponding RGB image and depth map. The best matching camera pose $\mathbf{T}_{\text{best}} \in \mathbb{R}^{3 \times 4}$ is determined by minimizing the mean Euclidean distance between the camera coordinates of the entire scene:

$$\mathbf{T}_{\text{best}} = \arg\min_{\mathbf{T}_i} \left( \frac{1}{N} \sum_{k=1}^{N} \|\boldsymbol{p}_k(\mathbf{T}_{\text{target}}) - \boldsymbol{p}_k(\mathbf{T}_i)\|_2 \right)$$

---

[1]Examples are shown in Appendix A.1

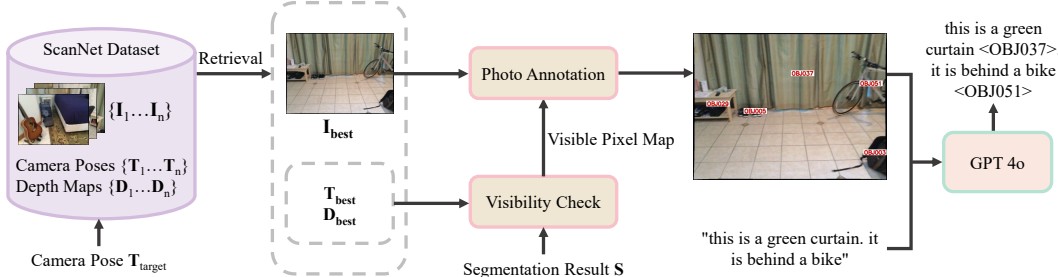

Figure 4: DetailedScanRefer generation pipeline: Given a ScanRefer description, we first retrieve its corresponding camera pose, $\mathbf{T}_{\text{target}}$. Using a camera pose matching algorithm, we find the closest match, $\mathbf{T}_{\text{best}}$, from the ScanNet dataset, along with the corresponding image $\mathbf{I}_{\text{best}}$ and depth map $\mathbf{D}_{\text{best}}$. The semantic segmentation result $\mathbf{S}$ is then projected from 3D space onto the image using $\mathbf{T}_{\text{best}}$ and the intrinsic matrix of the scene. $\mathbf{D}_{\text{best}}$ is applied to filter visible pixels for each object, and the visible object IDs are annotated on $\mathbf{I}_{\text{best}}$. Finally, GPT-4o is used to append object ID to each object in the description.

where $N$ is the total number of points in the scene, $\mathbf{T}_{\text{target}} \in \mathbb{R}^{3\times4}$ is the camera pose corresponding to the ScanRefer description, and $\mathbf{T}_i \in \mathbb{R}^{3\times4}$ is the $i$-th candidate pose from the same scene in ScanNet. The term $\mathbf{p}_k(\mathbf{T}) \in \mathbb{R}^3$ represents the camera coordinates of the $k$-th point in the scene transformed by the camera pose $\mathbf{T}$. The Euclidean distance between the transformed points under different camera poses is averaged over all points in the scene, ensuring that the selected camera pose closely matches the viewpoint described by the ScanRefer description.

**Visibility and Object Annotation**   The need to do a visibility check during object annotation arises from annotation noise. Simply projecting an object's center onto the image can lead to incorrect annotations, as hidden or partially visible objects may be mistakenly included. This becomes particularly problematic in later stages, such as querying GPT4o for high-quality responses, where accurately labeling only visible objects is crucial. To address this issue, we project the 3D instance segmentation mask to image space and compare it with the scene's depth map. We project 3D points onto the 2D image plane using the camera parameters as follows:

$$u = \frac{f_x X_{\text{c}}}{Z_{\text{c}}} + c_x, \quad v = \frac{f_y Y_{\text{c}}}{Z_{\text{c}}} + c_y$$

where $(X_{\text{c}}, Y_{\text{c}}, Z_{\text{c}})$ represent the 3D point in the camera coordinate system, and $f_x, f_y$ and $c_x, c_y$ denote the focal lengths and principal points of the camera, respectively. Visibility is confirmed by comparing the estimated depth $Z_{\text{c}}$ with the depth map $D_{\text{depth}}$, using the condition visible $= |Z_{\text{c}} - D_{\text{depth}}(u,v)| \leq \delta$, where $\delta$=0.1 meter accounts for minor discrepancies due to sensor noise. Appendix A.2 shows the pixel-level visibility masks and how objects are annotated.

**Photo Annotation**   We generate annotated images by overlaying unique object identifiers at the mean pixel coordinates of each object's mask. For example, an object with index 13 is labelled as "OBJ013" to clearly tag visible objects in the image. Examples of these annotations can be found in the bottom row of Fig.7. These annotated images are then sent to the GPT-4 API, along with the original ScanRefer description, for automatic generation of detailed annotations. In the generated descriptions, object IDs are inserted after the object references. As shown in Fig.4, for the original description: "This is a brown guitar. It is leaning against the wall." The enhanced output is: "This is a brown guitar <**OBJ018**>. It is leaning against the wall <**OBJ032**>." Details of the prompts can be found in Appendix A.3.

**Data Cleaning and Quality Rating**   To ensure high-quality annotations, we implemented several data cleaning processes. Key steps included discarding annotations where the first object ID did not match the ground truth, in line with ScanRefer's assumption that the target object is described first. Additionally, any outputs containing NaN values were removed. The cleaned annotations were then used as ground truth for training, where the model predicted object IDs for each mentioned

object. Detailed statistics for each data cleaning step are provided in Appendix A.4. Furthermore, annotation quality evaluation details using GPT-4o are presented in Appendix A.5.

# 5 EXPERIMENTS

## 5.1 EXPERIMENTAL SETUP

**Training Data** We follow exactly the same training data setup as Chat3D-v2 (Wang et al., 2023b) so that our results are directly comparable. The training datasets include ScanRefer (Chen et al., 2020), Scan2Cap (Chen et al., 2021), ScanQA (Azuma et al., 2022), SQA3D (Ma et al., 2023), Multi3DRef (Zhang et al., 2023b), and NR3D (Achlioptas et al., 2020). We also use ObjAlign, which is a dataset for aligning object IDs with objects[2]. Among these datasets, only ScanRefer and Scan2Cap tasks use viewpoint information, as they are the only datasets providing it. The proposed DetailedScanRefer, with about 16,000 samples in total, is also used where specified. For validation, we use ScanRefer, Scan2Cap, ScanQA, SQA3D, and Multi3DRef to select the best model checkpoint. The Scan2Cap dataset is modified by associating a camera pose with each caption. We refer to Appendix B for details.

**Model and Training Specifications** The proposed GeLVM uses the Vicuna-7B-v1.5 as the LLM backbone which is fine-tuned using the Low-Rank Adaptation (LoRA) (Hu et al., 2022). To extract object features, we utilize the pretrained Uni3D (Chen et al., 2023a) as the 3D encoder which is frozen during training. The segmentation model Mask3D (Schult et al., 2023) is employed for consistent and accurate segmentation of the 3D data, which is also frozen during training. We have used code from OpenIns3D (Huang et al., 2024b) to assist with visibility checks and to develop visualization tools for pixel-level masks. The VCPE module uses 256-dim final positional embedding, and 128-dim Fourier features. The positional embedding projection layer and the multi-head attention module are also trainable components. There are 100 object proposals for each scene. The entire training process using Adam optimizer and cosine learning rate scheduler for 3 epochs requires approximately 11 hours on 4 NVIDIA A100 GPUs. Our training settings are as follows.

**Evaluation Metrics** For 3D grounding tasks, grounding accuracy is measured at two Intersections over Union (IoU) thresholds: 25% and 50%, referred to as Acc@0.25 and Acc@0.50, respectively. For language tasks, metrics such as BLEU scores (Papineni et al., 2002), METEOR (Lavie & Agarwal, 2007) and CIDEr (Vedantam et al., 2015) are used to measure the degree of overlap between the generated answer and the reference, with higher scores indicating better performance.

## 5.2 EXPERIMENTAL RESULTS

Table 1: Accuracy on ScanRefer (Chen et al., 2020) validation set using GeVLM at 0.25 and 0.50 IoU. Unique subset contains samples where the grounding object is unique in the scene, in contrast to Multiple where there are multiple objects of the same kind as the grounding object.

| System | Unique | | Multiple | | Overall | |
|---|---|---|---|---|---|---|
| | Acc@0.25 | Acc@0.50 | Acc@0.25 | Acc@0.50 | Acc@0.25 | Acc@0.50 |
| 3DJCG (Cai et al., 2022) | - | 64.3 | - | 30.8 | - | 37.3 |
| D3Net (Chen et al., 2022a) | - | 72.0 | - | 30.1 | - | 37.9 |
| 3D-LLM (BLIP2-flant5) (Hong et al., 2023) | - | - | - | - | 30.3 | - |
| Chat-3D v2* (Huang et al., 2023b) | 79.0 | 74.5 | 34.7 | 31.6 | 42.9 | 39.6 |
| GeVLM (Ours) | **82.0** | **75.7** | **39.0** | **34.7** | **46.9** | **42.3** |

**Results on 3D Grounding Tasks** We first show the 3D grounding performance using ScanRefer and Multi3DRefer datasets. On the ScanRefer dataset shown in Tab.1, the proposed GeVLM achieved consistent performance improvement compared to the Chat-3D v2 baseline with clear margins for both Unique and Multiple subsets. The improvement is particularly pronounced when there are multiple confusing objects with similar semantic classes in the scene, demonstrating the importance of viewpoint and relative distance information which are crucial to distinguishing those objects. Overall, GeVLM achieved a 4.0% absolute accuracy improvement at 0.25 IoU and a 2.7% improvement at 0.50 IoU respectively compared to the Chat-3D v2 baseline.

---

[2]Questions are in the format of "what is Obj14" and the answer is "chair".

Table 2: F1 scores at 0.5 IoU on Multi3DRefer (Zhang et al., 2023b) validation set. ZT, ST, and MT refer to zero, single, and multiple target objects in the scene referenced by each description. "D" refers to distracting objects of the same semantic class.

| | ZT w/o D | ZT w/ D | ST w/o D | ST w/ D | MT | All |
|---|---|---|---|---|---|---|
| Chat-3D v2* (Huang et al., 2023b) | **90.7** | 62.2 | 64.3 | 33.0 | **42.1** | 44.9 |
| GeVLM (Ours) | 90.3 | **68.0** | **68.0** | **36.5** | 36.7 | **46.1** |

Table 4: Ablation study on ScanRefer validation set with VCPE, DetailedScanrefer (Detailed), and DACE. In the table, "World", "Camera", and "Rotate" refer to world coordinates, camera coordinates (both rotation and translation), and rotated coordinates (rotation only), respectively.

| Method | VCPE | Detailed | DACE | Unique | | Multiple | | Overall | |
|---|---|---|---|---|---|---|---|---|---|
| | | | | Acc@0.25 | Acc@0.50 | Acc@0.25 | Acc@0.50 | Acc@0.25 | Acc@0.50 |
| Chat-3D v2* | – | – | – | 79.0 | 74.5 | 34.7 | 31.6 | 42.9 | 39.6 |
| Ours | World | – | – | 81.1 | **76.1** | 35.8 | 32.0 | 44.2 | 40.2 |
| | Camera | – | – | 79.0 | 74.1 | 35.6 | 32.2 | 43.6 | 40.0 |
| | Rotate | – | – | 79.6 | 74.7 | 36.2 | 32.6 | 44.2 | 40.4 |
| | Rotate | ✓ | – | 80.7 | 74.9 | 35.7 | 32.3 | 44.0 | 40.2 |
| | Rotate | – | $T=0.05$ | 79.5 | 73.7 | 37.9 | 33.7 | 45.6 | 41.1 |
| | Rotate | ✓ | $T=0.05$ | 80.4 | 74.1 | 38.1 | 34.0 | 46.0 | 41.4 |
| | Rotate | ✓ | $T=0.03$ | **82.0** | 75.7 | **39.0** | **34.7** | **46.9** | **42.3** |

We then extend our experiments to the Multi3DRefer dataset as shown in Tab.2 where an overall 1.2% absolute F1 score improvement is achieved. In particular, large improvements are found when there are semantically distracting classes, with a 5.8% absolute F1 score improvement on the zero target subset (i.e. the target object is not in the scene) and 3.5% on the single target subset when distractors are added. For the MT subset, where multiple objects sharing the same semantics need to be grounded, we observe a 5.4% performance drop compared to the baseline method. This can be attributed to the nature of the task. First, grounding multiple objects requires less spatial reasoning, making our proposed VCPE less effective. Second, since the task involves grounding multiple objects that share the same semantics, the model relies more heavily on object category recognition than on spatial differentiation, further diminishing the effectiveness of the DACE loss.

Table 3: Results on language tasks using ScanQA validation set and SQA3D test set. B1 to B4 represents BLEU-1 to 4, M for METEOR, C for CIDEr, and R for ROUGE-L. The Chat-3D v2 is the reproduced results which is slightly better than the reported numbers in the original paper.

| System | ScanQA | | | | | | | | SQA3D | | | | | | |
|---|---|---|---|---|---|---|---|---|---|---|---|---|---|---|---|
| | B1 | B2 | B3 | B4 | M | C | R | EM | What | Is | How | Can | Which | Others | Avg |
| LLaVA (zero-shot) (Liu et al., 2023) | 7.1 | 2.6 | 0.9 | 0.3 | 10.5 | 5.7 | 12.3 | 0.0 | – | – | – | – | – | – | – |
| LL3DA (Chen et al., 2024a) | – | – | – | 13.5 | 15.9 | 76.8 | 37.3 | – | – | – | – | – | – | – | – |
| LEO (Huang et al., 2024a) | – | – | – | 13.2 | **20.0** | **101.4** | **49.2** | 24.5 | – | – | – | – | – | – | 50.0 |
| Scene-LLM (Fu et al., 2024) | 42.2 | 26.4 | 18.7 | 11.7 | 15.8 | 80.0 | 35.9 | **25.6** | 40.0 | **69.2** | 42.8 | **70.8** | **46.6** | 52.5 | **53.6** |
| 3D-LLM (BLIP2-flant5) (Hong et al., 2023) | 39.3 | 25.2 | 18.4 | 12.0 | 14.5 | 69.4 | 35.7 | 20.5 | – | – | – | – | – | – | – |
| Chat-3D v2* (Huang et al., 2023b) | 42.3 | 28.1 | 19.6 | 13.4 | 18.0 | 88.9 | 42.1 | 22.4 | 43.9 | 66.0 | **52.5** | 66.3 | 46.4 | 50.2 | 52.5 |
| GeVLM (Ours) | **42.4** | **28.7** | **21.3** | **15.4** | 18.1 | 90.5 | 41.8 | 21.7 | **44.1** | 68.6 | 52.3 | 62.7 | 45.6 | **55.8** | 53.5 |

**Results on Language Tasks: 3D QA and Captioning**  In addition to object grounding, GeVLM is also beneficial in language tasks, as shown in Table 3. While not explicitly designed to enhance language tasks, GeVLM achieves the best performance across most metrics compared to other 3D LLMs capable of 3D grounding tasks on ScanQA and achieved on-par performance with Chat-3D v2. Notably, among the listed models, only 3D-LLM, Chat-3D v2, and GeVLM possess grounding capabilities, emphasizing our model's superior versatility and performance in both grounding and language-based tasks. Detailed Scan2Cap results are provided in Appendix B.

**Ablation Studies**  Ablation studies are performed using the ScanRefer dataset to better understand the effect of each proposed component in GeVLM, as shown in Tab.4.

Table 5: Comparison of predicted vs. target object center distances on Scanrefer, Both average and median distances are reported for unique and multiple scenarios, with and without DACE loss.

| System | Unique | | Multiple | |
|---|---|---|---|---|
| | Mean | Median | Mean | Median |
| VCPE(r) | 0.60 | 0.04 | 1.61 | 1.13 |
| VCPE(r) + DACE | **0.52** | **0.04** | **1.49** | **0.91** |

The world, camera, and rotated coordinate systems were analyzed using the same VCPE model. The rotated coordinates showed the best performance. VCPE is especially useful in the Multiple case, where it helps clarify which object is being referenced among similar objects by focusing on their relative positions. Accuracy improves further in the Multiple case when using the DACE loss, which emphasizes spatial distance in the label space, rather than relying only on semantic similarity. In the Unique case, using DetailedScanrefer alone improves performance, but the best results are achieved when combining DetailedScanrefer with DACE loss and VCPE.

Moreover, to quantitatively demonstrate the effect of the DACE loss, the average (mean and median) distance between the predicted and the target object centers across all test samples is presented in Tab.5. The significantly smaller average distances observed in both scenarios indicate that the DACE loss helps GeVLM focus on locations that are spatially closer to the target object.

**Qualitative Analysis**   We demonstrate the advantages of GeVLM through examples in Fig. 5, highlighting four common types of spatial confusion: left/right, near/far with respect to the camera, front/back, and geographical directions. In cases (A) to (C), GeVLM, equipped with VCPE and DACE, accurately selects the correct object based on the description, effectively resolving ambiguities that the baseline model fails to address. The baseline consistently picks objects with the correct semantic category but incorrect spatial positioning. In case (D), although GeVLM's prediction does not perfectly overlap with the ground truth bounding box, it aligns with the described location, indicating a better understanding of the spatial context compared to the baseline, which selects an object without considering positional cues.

## 6 CONCLUSION

This paper introduces a geometry-enhanced visual language model (GeVLM) to improve 3D object grounding and scene understanding. By integrating viewpoint-consistent position encoding (VCPE) and distance-aware cross-entropy (DACE) loss, GeVLM achieves improved interpretation of spatial relationships, while effectively incorporating distance information into the label space. Additionally, the DetailedScanRefer dataset is proposed to offer dense object identifiers that complement object reference descriptions, enhancing the model's spatial reasoning capabilities. GeVLM achieves significant performance gains over the Chat3D-v2 baseline, with notable improvements of 4.0% in Acc@0.25 and 2.7% in Acc@0.50 on the ScanRefer benchmark.

## 7 LIMITATIONS

Our work has several limitations, including the need for further refinement of the camera pose-matching algorithm to achieve optimal performance. Our approach is specifically tailored to tasks that rely heavily on viewpoint information, making accurate camera pose information essential to fully leverage the model's capabilities. Many existing datasets lack this type of annotation, limiting the applicability of our method. Furthermore, the prompt design for photo annotation could be improved to enhance both efficiency and precision. Lastly, while this paper uses the Chat-3D-v2 as the baseline and follows the exact model and training configuration for direct comparability, we also notice that, as a fast-evolving field, the latest work, such as Chat-Scene (Huang et al., 2023a), has proposed foundation models that surpass the performance of Chat-3D-v2 baseline by clear margins. However, our proposed methods are orthogonal to these advancements and, in theory, could be applied to achieve further improvements. Exploring these opportunities will be important future work when additional resources and time are available.

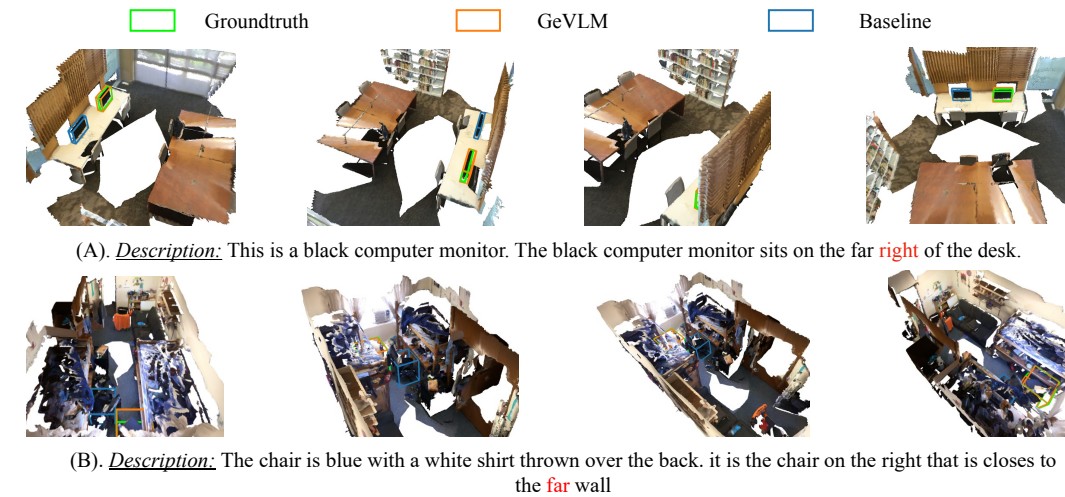

(A). *Description:* This is a black computer monitor. The black computer monitor sits on the far right of the desk.

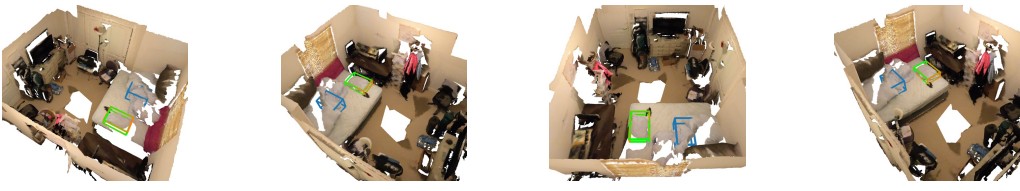

(B). *Description:* The chair is blue with a white shirt thrown over the back. it is the chair on the right that is closes to the far wall

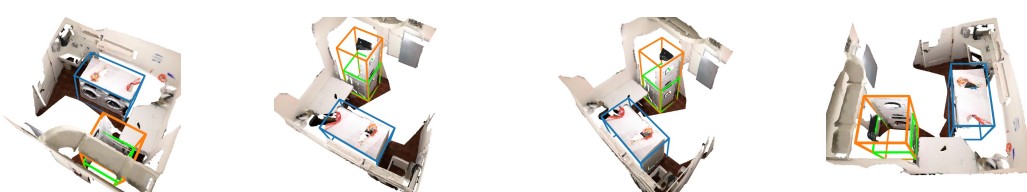

(C). *Description:* There is a light grey pillow on the bed. it is smaller than the other pillows and in front of the red pillow.

(D). *Description:* The clothes dryer is in the northeast corner of the room. the clothes dryer has a white color and a half circle mirror in the center

Figure 5: Comparison between GeVLM and baseline (Chat-3D v2) on viewpoint-related examples with potential ambiguities, including: (A) left/right, (B) near/far, (C) front/back and (D) north/south/east/west. The description is associated with a specific viewpoint and hence becomes confusing in other viewpoints.

## 8 REPRODUCIBILITY STATEMENT

We have made all the code, model checkpoints, and DetailedScanrefer used in this work available at https://anonymous.4open.science/r/GeVLM-1372/.

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

# A  ANNOTATION PROCEDURE FOR DETAILEDSCANREFER

## A.1  COMPARISON BETWEEN RENDERED IMAGE AND REAL-WORLD PHOTO

We explored the use of rendered images for annotation but found the image quality lacking compared to real-world photos retrieved from the Scannet dataset using the Camera Pose Matching algorithm. The rendered images often suffer from poor lighting, texture quality, and geometric accuracy, making them less suitable for precise annotations. By comparing these renderings with ScanNet photos, the limitations of synthetic data are clear, highlighting the need for higher-fidelity imagery or real-world data for accurate annotation tasks.

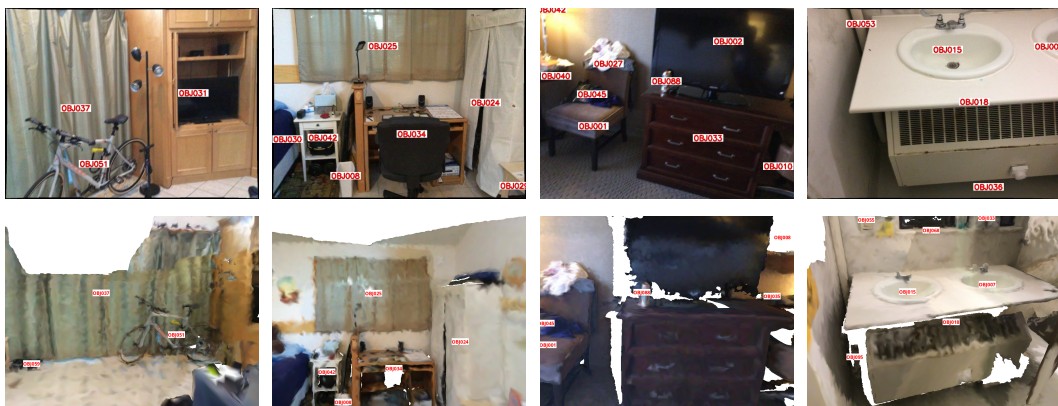

Figure 6: Top row: Scannet Photos. Bottom row: Corresponding Rendered Images. Each pair of images corresponds to the same ScanRefer description.

## A.2  EXAMPLE OF PIXEL-LEVEL VISIBILITY MASKS

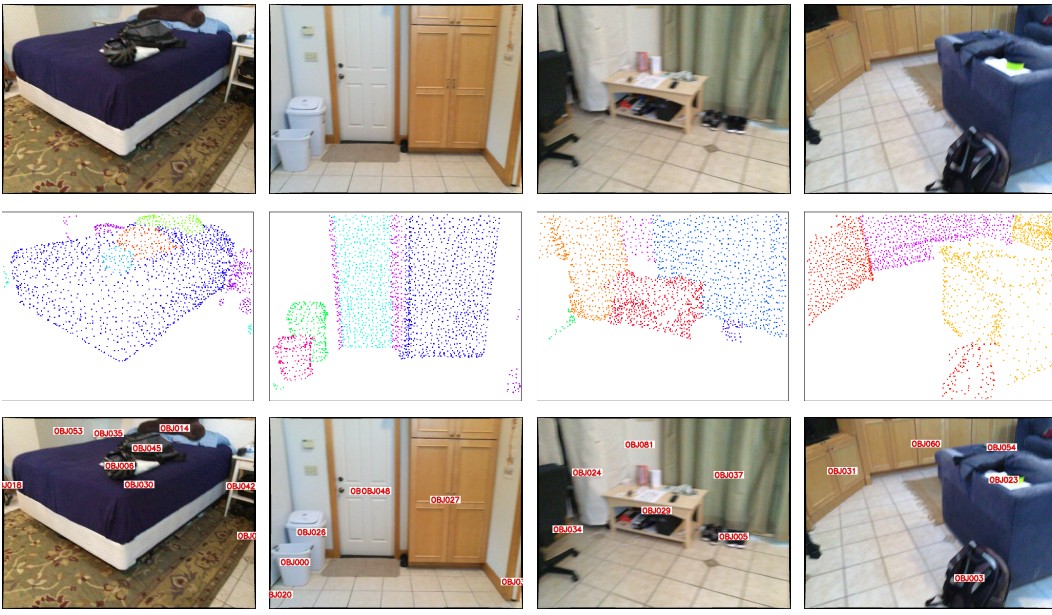

Figure 7: Pixel-level segmentation masks for object visibility checks. Top: original photos. Middle: corresponding segmentation masks. Bottom: annotated photos with object identifiers.

## A.3 GPT-4O ANNOTATION PROMPT

This is the prompt we used for GPT annotation:

> "You are a helpful assistant designed to output JSON. The task is to identify all mentioned objects in the image and add the matching obj id to the given description. The OBJID is shown in red font, and it should be annotated at the centre of the object. Remember, please return both the <input_description> and the <augmented_description> with obj id added. You should not modify the <input_description>. Only add the <OBJID> after the object entity if you can recognize both the object and the red annotation clearly in the image. Also, if you cannot recognize ALL of the objects AND ALL of their corresponding red annotation in the description, simply output "NAN" in the "augmented_description". An example is here: "input_description: This is a brown chair. it is at a high table. augmented_description: This is a brown chair <OBJ003>. it is at a high table <OBJ012>."

This is the prompt format we used for the DetailedScanrefer Dataset:

> "According to the provided description, <input_description>, please append the correct object ID after each object mentioned in the description."

The <input_description> refers to the original ScanRefer description. The annotations generated by GPT-4o serve as the reference captions for each corresponding question.

## A.4 DATASET STATISTICS

We provide the dataset statistics in Tab.6. The numbers of descriptions before and after each processing step are shown.

| Description | Count |
|---|---|
| Before processing | 32,338 |
| Inconsistent first ObjId | 13,836 |
| NaN values | 2,191 |
| No ObjId | 154 |
| Invalid ObjId range | 6 |
| **After processing** | 16,151 |

Table 6: Data Processing Statistics

| Rating | Count | Percentage |
|---|---|---|
| 1 | 28 | 0.09% |
| 2 | 3291 | 10.18% |
| 3 | 7092 | 21.93% |
| 4 | 5617 | 17.38% |
| 5 | 16310 | 50.42% |
| **Total** | 32338 | 100% |

Table 7: Distribution of GPT-4o Ratings

## A.5 DATASET QUALITY EVALUATION

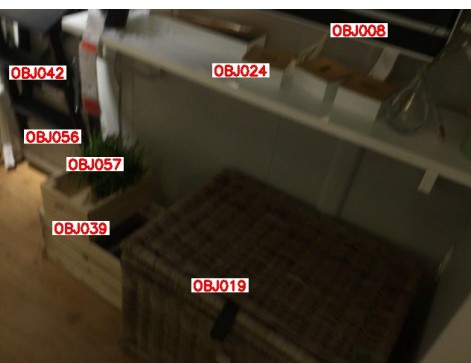 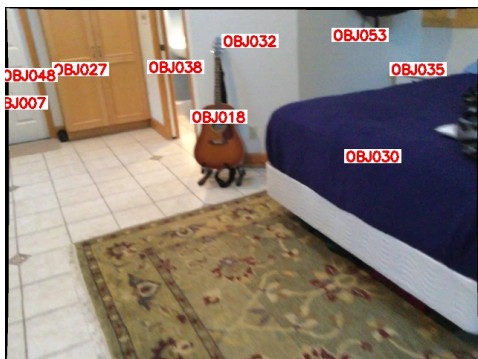

**GPT Annotation:** "this is a wicker cabinet <OBJ019>. it is to the right of the boxes <OBJ039>."

**Rating:** 5/5

**Reasoning:** The annotations are correct. The wicker cabinet labeled as <OBJ019> matches the object in the image. The boxes labeled <OBJ039> are also correctly identified and are to the left of the wicker cabinet, which aligns with the described spatial relationship.

**GPT Annotation:** "this is a brown cabinet <OBJ027>. it is to the right of a picture <OBJ032>"

**Rating:** 1/5

**Reasoning:** The object IDs are incorrectly attributed in the description. The brown cabinet is identified as <OBJ027>, which matches the visible cabinet in the image. However, the picture is labeled as <OBJ032> in the image, but spatially it does not appear to be to the left of the cabinet. Additionally, the context does not logically support the described spatial relationship.

Figure 8: Comparison of two rating scenarios in different scenes.

We also evaluate the annotation quality automatically using GPT4o with the following prompt:

> "You are tasked with evaluating the accuracy and completeness of text annotations provided for objects in an image. Some objects in the image is labeled with an object ID (e.g., <OBJ014>), and these IDs are referenced in the text annotations. Your goal is to ensure that every object mentioned in the text annotation has a accurate corresponding red-text annotation in the image. First, verify that all objects mentioned in the text are annotated in the image. Second, ensure that the object descriptions in the text correctly match the labeled objects in the image in terms of type, appearance, and location. After reviewing, provide a rating between 1 and 5, where 1 represents poor annotation quality and 5 represents excellent quality. The rating should consider whether all objects mentioned in the text are annotated in the image, and whether the descriptions are accurate."

We submitted both the annotated photo and its detailed description to GPT-4o, requesting it to rate their consistency and accuracy on a scale from 1 to 5. An average rating of 3.31 was achieved across 32,338 descriptions. Specifically, the distribution of ratings is presented in Tab.7. Two examples are shown in Fig.8, where the annotation, rating, and its reasoning are illustrated.

# B ADDITIONAL RESULTS ON SCAN2CAP

In this appendix, we provide a detailed explanation of the modifications made to the Scan2Cap dataset for our viewpoint-aware captioning task. We also discuss the implications for evaluation and how these changes affect comparability with existing models.

**Scan2Cap** (Chen et al., 2021) is a captioning dataset generated based on **ScanRefer** (Chen et al., 2020), which provides natural language descriptions of objects within 3D indoor scenes from the ScanNet dataset (Dai et al., 2017).

Each description in ScanRefer is associated with a specific camera pose. By reusing these camera poses, we reconstruct viewpoints for the Scan2Cap dataset, making it **viewpoint-aware**. This approach enhances the dataset by incorporating spatial context and specific viewpoints, providing a more comprehensive captioning task.

The Scan2Cap dataset utilizes a set of predefined prompts to guide the captioning task. Notably, these prompts are used in the original Chat-3D-v2 task (Huang et al., 2023b). The prompts are designed to elicit detailed descriptions of objects and their spatial relationships within a scene. The prompts include:

1. "Begin by detailing the visual aspects of the <id> before delving into its spatial context among other elements within the scene."

2. "First, depict the physical characteristics of the <id>, followed by its placement and interactions within the surrounding environment."

3. "Describe the appearance of the <id>, then elaborate on its positioning relative to other objects in the scene."

4. "Paint a picture of the visual attributes of <id>, then explore how it relates spatially to other elements in the scene."

5. "Start by articulating the outward features of the <id>, then transition into its spatial alignment within the broader scene."

6. "Provide a detailed description of the appearance of <id> before analyzing its spatial connections with other elements in the scene."

7. "Capture the essence of the appearance of <id>, then analyze its spatial relationships within the scene's context."

8. "Detail the physical characteristics of the <id> and subsequently examine its spatial dynamics amidst other objects in the scene."

9. "Describe the visual traits of <id> first, then elucidate its spatial arrangements in relation to neighboring elements."

10. "Begin by outlining the appearance of <id>, then proceed to illustrate its spatial orientation within the scene alongside other objects."

An example entry from the original Scan2Cap dataset is provided below. Multiple reference captions correspond to a single prompt, offering varied descriptions of the object.

> **Prompt:** First, depict the physical characteristics of the <OBJ014>, followed by its placement and interactions within the surrounding environment.

The corresponding reference captions are as follows:

- There are brown wooden cabinets. Placed on the side of the kitchen.
- There is a set of bottom kitchen cabinets in the room. It has a microwave in the middle of it.
- There is a set of bottom kitchen cabinets in the room. There is a microwave in the middle of them.
- Brown kitchen cabinets, the top is decorated with marble layers, and it is placed on the left in the direction of view. On the right, there are four brown chairs.
- The kitchen cabinets are located along the right wall. They are below the countertop. The kitchen cabinets are located to the right of the table and chairs.

In our modified viewpoint-aware Scan2Cap dataset, each question is associated with a specific viewpoint (camera pose). Each viewpoint corresponds to one correct reference caption. The dataset includes camera parameters (`position`, `rotation`, `lookat`) for each entry.

Example entries:

> **Prompt:** "First, depict the physical characteristics of the <OBJ014>, followed
> by its placement and interactions within the surrounding environment.",

Given a particular camera pose it would have one reference caption corresponding to the origin
Scanrefer Dataset:

- "There is a set of bottom kitchen cabinets in the room. There is a microwave in the middle
  of them."

The modification of the dataset has significant implications for evaluation: The evaluation method
changes due to the dataset modification. Each question under a specific viewpoint has a single
correct reference caption. This differs from the original method, which averaged over multiple
reference captions. Consequently, the modified dataset and evaluation method are **not directly
comparable** to models trained on the original Scan2Cap dataset.

Table 8: Evaluation results on Scan2Cap validation set at IoU thresholds 0.25 and 0.50.

| System | @0.25 IoU | | | @0.5 IoU | | |
| --- | --- | --- | --- | --- | --- | --- |
| | CIDER | BLEU-4 | METEOR | CIDER | BLEU-4 | METEOR |
| Chat-3D-v2* (Huang et al., 2023b) | **72.20** | **11.28** | 18.89 | **68.63** | **10.46** | 18.23 |
| Ours | 67.94 | 11.19 | **19.06** | 64.47 | 10.44 | **18.38** |

## C  COMPARISON OF GEVLM AND EXPERT MODELS

| Category | System | Unique | | Multiple | | Overall | |
|---|---|---|---|---|---|---|---|
| | | Acc@0.25 | Acc@0.5 | Acc@0.25 | Acc@0.5 | Acc@0.25 | Acc@0.5 |
| Expert Model | ScanRefer (Chen et al., 2020) | 76.33 | 53.51 | 32.73 | 21.11 | 41.19 | 27.40 |
| | MVT (Huang et al., 2022) | 77.67 | 66.45 | 31.92 | 23.30 | 39.43 | 33.26 |
| | 3D-SPS (Luo et al., 2022) | 84.12 | 66.72 | 40.32 | 29.82 | 48.36 | 36.98 |
| | ViL3DRel (Chen et al., 2022c) | 81.58 | 68.62 | 40.30 | 30.71 | 47.94 | 37.73 |
| | BUTD-DETR (Jain et al., 2022) | 84.20 | 66.30 | 46.60 | 35.10 | 52.20 | 39.80 |
| | HAM (Chen et al., 2022b) | 79.24 | 67.86 | 41.46 | 34.03 | 48.79 | 40.60 |
| | 3DRP-Net (Wang et al., 2023a) | 83.13 | 67.74 | 42.14 | 31.95 | 50.10 | 38.90 |
| | EDA (Wu et al., 2023) | 85.76 | 68.57 | **49.13** | 37.64 | 54.59 | 42.26 |
| | M3DRef-CLIP (Zhang et al., 2023b) | 85.30 | 77.20 | 43.80 | 36.80 | 51.90 | 44.70 |
| | ConcreteNet (Unal et al., 2024) | **86.40** | **82.05** | 42.41 | 38.39 | 50.61 | **46.53** |
| | DORa (Wu et al., 2024) | - | - | - | - | **52.80** | 44.80 |
| Fine-tuned General Backbone | 3D-VLP (Jin et al., 2023) | 84.23 | 64.61 | 43.51 | 33.41 | 51.41 | 39.46 |
| | 3D-VisTA (Ziyu et al., 2023) | 81.60 | 75.10 | 43.70 | **39.10** | 50.60 | 45.80 |
| 3D Grounding + 3D Captioning | 3DJCG (Cai et al., 2022) | 83.47 | 64.34 | 41.39 | 30.82 | 49.56 | 37.33 |
| | D3Net (Chen et al., 2022a) | - | 72.04 | - | 30.05 | - | 37.87 |
| 3D Grounding + 3D Captioning +3D Q&A | GeVLM (Ours) | 82.00 | 75.70 | 39.00 | 34.70 | 46.90 | 42.30 |

Table 9: Performance comparison of models on Scanrefer.

## D  QUALITATIVE RESULT OF MASK3D

Thhe table summarizing key statistics and metrics for Mask3D's performance across both the training and validation splits, including counts for $IoU \geq 0.25$ and $IoU \geq 0.50$, along with maximum IoU rates, to further demonstrate the quality and comprehensiveness of the generated proposals.

| Metric | Train Split Count | Validation Split Count |
|---|---|---|
| Total Count (Original ScanRefer Dataset) | 36,665 | 9,508 |
| $IoU \geq 0.25$ Count | 36,187 | 8,924 |
| $IoU \geq 0.50$ Count | 35,061 | 8,168 |
| Max IoU@0.25 | 0.9870 | 0.9386 |
| Max IoU@0.50 | 0.9563 | 0.8591 |

Table 10: Summary of Mask3D-generated dataset metrics for training and validation splits.

In the training process, we only use 32,338 annotations that meet the strict criterion of $IoU \geq 0.75$ with ground truth objects, ensuring that only highly accurate object proposals are retained. This high threshold reflects the precision and relevance of the dataset for effective downstream tasks.

