# OpenReview forum: "GeVLM: 3D Object Grounding with Geometry-enhanced Vision Language Model"
_ICLR.cc/2025/Conference — ICLR 2025 Conference Withdrawn Submission_

### Official Review · Reviewer_YvFj · 2024-10-19

**Soundness:** 3
**Presentation:** 3
**Contribution:** 2
**Rating:** 3
**Confidence:** 5

**Summary:**

The paper aims to solve a challenging 3D object grounding task. The paper proposes GeVLM to grasp view-point information and the relative spatial distance between objects. Besides, to facilitate the research community, the paper introduces a new dataset named DetailedScanRefer, which provides identifiers and spatial annotation for each object mentioned in the referencing description to further emphasize spatial relationships.

**Strengths:**

1. Focusing on positional relationships is a very worthy research problem for 3D tasks.
2. Experiments validate the performance of the proposed method by comparing some advanced algs.

**Weaknesses:**

1. The way to integrate view-point information seems too incremental and has limited novelty. It simply utilize position embedding then use self-attention.
2. The gain of VCPE and the designed DetailedScanRefer dataset in the ablation study are too small to prove the effectiveness of the method.
3. The paper lacks qualitative experimental analysis to prove that VCPE and DACE have learned the expected position information.

**Questions:**

See weaknesses

---

> ### Author Response · Authors · 2024-11-21
> **Response to Reviewer YvFj**
>
> >The way to integrate view-point information seems too incremental and has limited novelty. It simply utilize position embedding then use self-attention.
>
> While we utlize standard modules such as *position embeddings* and *self-attention* mechanisms, the key innovation lies in the position encoding itself. Unlike traditional positional encodings, VCPE incorporates rotation transformations to condition positional information based on the observer's viewpoint.
>
> This representation has not been explored in prior work and offers a theoretically sound method for capturing spatial relationships from various perspectives. By transforming object coordinates according to the observer's viewpoint, our approach ensures robustness to translations, allowing the model to better understand spatial relationships.
>
> >The gain of VCPE and the designed DetailedScanRefer dataset in the ablation study are too small to prove the effectiveness of the method.
>
> We respectfully disagree with the assessment that the gains are too small to demonstrate the effectiveness of our approach. As shown in our ablation study (see Table below), incorporating VCPE and DACE consistently yields meaningful improvements across multiple metrics.
>
> - **Overall Performance Gains**: Our best model achieves an Overall Acc\@0.25 of 46.9% and Acc\@0.50 of 42.3%, marking significant improvements over the baseline Chat-3D v2* (42.9% and 39.6%, respectively).
>
> - **Unique and Multiple Targets**: The improvements are consistent across both the Unique and Multiple target subsets. Notably, in the Multiple Targets subset, we observe an increase in Acc\@0.25 from 34.7% to 39.0%, which represents a substantial gain for this challenging task.
>
> In the context of 3D grounding, even modest percentage improvements are highly significant due to the inherent complexity of the task. The consistent performance gains across various settings demonstrate the effectiveness and practical value of our method.
>
> >The paper lacks qualitative experimental analysis to prove that VCPE and DACE have learned the expected position information.
>
> We would like to highlight that we have already provided several qualitative examples in the paper. Specifically, **Figure 3** and **Figure 6** demonstrate how VCPE and DACE contribute to learning and conveying the expected positional information. Additionally, we have dedicated a subsection in **Section 5.2** to qualitative anlysis, which was originally titled 'Qualitative Examples' and has now been revised to 'Qualitative Analysis' for clarity.
>
> We recognize the importance of qualitative analysis in supporting our findings, and we will ensure that these sections are prominently emphasized in the revised version of the paper for greater clarity and visibility.

---

### Official Review · Reviewer_1oXE · 2024-10-31

**Soundness:** 2
**Presentation:** 1
**Contribution:** 2
**Rating:** 5
**Confidence:** 4

**Summary:**

This paper points out that many 3D LLM models often do not consider the viewpoint information and the relative spatial distance between objects. It introduces a method for 3D scene understanding using LLM that mainly focuses on incorporating 3D viewpoint information, improving positional encoding, implementing distance-aware cross-entropy loss, and enhancing the dataset's quality. The model trained using the proposed method improves upon the baseline by clear margins.

**Strengths:**

1. This paper notices an interesting aspect that other 3D LLM may overlook. The corresponding method is specifically tailored to tasks and successfully tackles the identified problems, as shown by the ablation studies.
2. The proposed method is orthogonal to the existing methods and can be used to improve other methods.

**Weaknesses:**

1. The improvement on the Scan2Cap dataset is weak, although the author incorporates extra information for captioning, and the proposed method may have a negative impact. Some recent baselines, for example, LEO[1] and LL3DA[2], are not compared. In line 453, the caption ability is mentioned, but no results are provided.
2. The improvement on SQA, a benchmark focusing on viewpoint, is weak. Since this paper mainly focuses on improving the view understanding of the LLM, the results are not compelling enough.
3. In line 418, 'the task prioritizes object semantics over spatial location, further diminishing the effectiveness of the DACE loss.' the paper does not give any evidence to support this claim.
4. Although the method surpasses the baseline, it also uses extra annotations. No ablations can be used to decide which aspect contributes to the performance gain. For example, the setting in which only VCPE DACE is used and no detailed ScanRefer annotations are added.

[1] https://arxiv.org/abs/2311.12871
[2] https://arxiv.org/abs/2311.18651

**Questions:**

1. Is the view information only used during training? Since we can not access the ground-truth view information, this hinders the wide application of the proposed method.
2. Some matrices, such as the ROUGE-L and EM on the ScanQA dataset, are not reported.
3. Why are "near" and "far" considered viewpoint-related questions with potential ambiguities in line 519? The distance between objects will remain constant from the given viewpoint.
4. Why is 'Chat-3D v2' marked with * in Tables 1, 3, and 4 but not in Table 2? Does this denote reproduced results?

---

> ### Author Response · Authors · 2024-11-21
> **Response to Reviewer 1oXE Part 1/3**
>
> We thank the reviewer for the constructive suggestions, and we would like to address the concerns as follows.
>
>
> **Weaknesses:**
>
> > Weakness 1: The improvement on the Scan2Cap dataset is weak, although the author incorporates extra information for captioning, and the proposed method may have a negative impact.
>
> We acknowledge the concern regarding the seemingly modest improvement on the Scan2Cap dataset. However, we'd like to clarify that our modifications have introduced a more challenging, viewpoint-aware captioning task, which significantly affects evaluation metrics and comparability with existing models.
>
> 1. **Viewpoint-Aware Augmentation**: We augmented the Scan2Cap dataset by reintroducing camera poses from the original ScanRefer dataset, making it viewpoint-aware. Each prompt is now associated with a specific viewpoint and corresponds to a single unique reference caption. This adds spatial context to the captioning task, requiring models to generate descriptions accurate from particular perspectives within the scene.
>
> 2. **Stricter Evaluation Protocol**: Our augmentation leads to a more stringent evaluation process. Unlike the original dataset, where evaluation metrics are averaged over multiple reference captions per prompt, we evaluate the model's output against a single, specific reference caption tied to a viewpoint. This makes direct comparison with models trained on the original Scan2Cap dataset inappropriate, as the tasks are fundamentally different in difficulty.
>
> 3. **Competitive Performance Despite Increased Difficulty**: Despite the heightened challenge, our model demonstrates competitive performance, highlighting the effectiveness of incorporating viewpoint information. We believe this advancement aligns more closely with real-world applications where an observer's perspective is crucial.
>
>
> We have provided detailed explanations of our dataset augmentation, evaluation implications, and results in **Appendix B** in our paper.
>
>
> >Some recent baselines, for example, LEO[1] and LL3DA[2], are not compared.
>
> Note that upon reviewing these works, we found that both LEO and LL3DA are not able to perform 3D grounding tasks.
>
> #### ScanQA Metrics
>
> | System     | B1     | B2     | B3     | B4     | M      | C      | R      | EM     |
> |------------|--------|--------|--------|--------|--------|--------|--------|--------|
> | LL3DA      | --     | --     | --     | 13.5   | 15.9   | 76.8   | 37.3   | --     |
> | LEO        | --     | --     | --     | 13.2   | 20.0   | 101.4  | 49.2   | 24.5   |
> | GeVLM      | 42.4   | 28.7   | 21.3   | 15.4   | 18.1   | 90.5   | 41.8   | 21.7   |
>
>
> #### SQA3D Metrics
>
> | System     | What   | Is     | How    | Can    | Which  | Others | Avg    |
> |------------|--------|--------|--------|--------|--------|--------|--------|
> | LEO        | --     | --     | --     | --     | --     | --     | 50.0   |
> | GeVLM      | 44.1   | 68.6   | 52.3   | 62.7   | 45.6   | 55.8   | 53.5   |
>
>
> Our model outperforms LL3DA on the ScanQA task, but LL3DA is not applicable to the SQA3D task. Regarding LEO, we outperform it on the SQA3D task but observe slightly lower performance on the ScanQA task for some metrics.  We have updated our paper to include comparisons with these models on ScanQA and SQA3d tasks in Table 3.
>
> >In line 453, the caption ability is mentioned, but no results are provided.
>
> As stated in line 362, we have provided results for Scan2Cap in the Appendix B. We included further clarification in Section 5.2 at the suggested place.

---

> ### Author Response · Authors · 2024-11-21
> **Response to Reviewer 1oXE Part 2/3**
>
> >The improvement on SQA, a benchmark focusing on viewpoint, is weak. Since this paper mainly focuses on improving the view understanding of the LLM, the results are not compelling enough.
>
> We would like to clarify the following points:
>
> - **Lack of Explicit Viewpoint Annotations in ScanQA**:
> The ScanQA dataset does not provide explicit viewpoint annotations or record camera poses during data collection. Annotators used an interactive 3D scene viewer to freely navigate and observe scenes from multiple angles without documenting specific viewpoints.
> In datasets like **ScanRefer**, each description is linked to a specific camera viewpoint. **SQA3D** explicitly states the viewpoint in the initial sentence of each description. It is infeasible to reconstruct or infer the annotators' viewpoints in ScanQA, unlike our approach with the **Scan2Cap** dataset.
>
> - **Limited Viewpoint-Related Content**:
> The dataset's documentation does not specify how many questions are viewpoint-related. Many questions focus on viewpoint-invariant attributes such as object color or quantity (e.g., "What color is the plastic clothes hanger?" or "How many armchairs are there?"). Incorporating viewpoint information in these cases does not provide additional benefits.
>
> - **Dataset Constraints Affecting Results**:
> The modest improvements observed are not indicative of our method's effectiveness but reflect the constraints imposed by the dataset itself. Our viewpoint-aware approach demonstrates benefits on datasets where viewpoint information is available.
>
>
> We hope this clarifies the performance results on ScanQA and underscores the effectiveness of our approach when appropriate viewpoint information is available.
>
> >In line 418, 'the task prioritizes object semantics over spatial location, further diminishing the effectiveness of the DACE loss.' the paper does not give any evidence to support this claim.
>
> - As noted in the caption of **Table 2** in our paper, the Multiple Targets (MT) task involves scenarios where a scene contains multiple instances of the target object. Unlike single-target tasks that require distinguishing a specific object based on spatial relationships to surrounding objects, the MT task focuses on identifying all instances of a target object within the scene. This shifts the emphasis toward object semantics derived from the language description (e.g., object type and attributes) rather than spatial location or proximity. Therefore, the task inherently prioritizes object semantics over spatial information, which explains why spatial differentiation mechanisms like the DACE loss may be less effective.
> - The DACE loss is defined in **Equations (4)** and **(5)** of our paper (please refer to these equations for detailed formulation). The DACE loss is designed to enhance fine-grained spatial differentiation by assigning higher importance to closer objects and penalizing distant ones. In the context of the MT task, where the goal is to recognize all relevant objects regardless of their spatial relationships, the spatial emphasis introduced by the DACE loss does not align with the task objectives. This misalignment can lead to diminished performance, as observed in our results in Table 2, because the loss function adds unnecessary spatial constraints to a task that primarily relies on object semantics.
>
> >Although the method surpasses the baseline, it also uses extra annotations. No ablations can be used to decide which aspect contributes to the performance gain. For example, the setting in which only VCPE DACE is used and no detailed ScanRefer annotations are added.
>
> We have added a comparison row for **VCPE + DACE** in the table below in the paper for clarity:
>
> | VCPE    | Detailed | DACE      | Unique (Acc\@0.25) | Unique (Acc\@0.50) | Multiple (Acc\@0.25) | Multiple (Acc\@0.50) | Overall (Acc\@0.25) | Overall (Acc\@0.50) |
> |---------|----------|-----------|-------------------|-------------------|---------------------|---------------------|-------------------|-------------------|
> | Rotate  | --       | --        | 79.6              | 74.7              | 36.2                | 32.6                | 44.2              | 40.4              |
> | Rotate  | --       | $T$=0.05  | 79.5              | 73.7              | 37.9                | 33.7                | 45.6              | 41.1              |
>
> As shown in the table, incorporating DACE without DetailedScanRefer results in a noticeable performance improvement across multiple metrics, including an increase in **Multiple Acc\@0.25**, **Multiple Acc\@0.50**, and overall metrics, demonstrating the effectiveness of DACE even in the absence of DetailedScanRefer.

---

> ### Author Response · Authors · 2024-11-21
> **Response to Reviewer 1oXE Part 3/3**
>
> **Questions:**
>
> >Question 1: Is the view information only used during training? Since we can not access the ground-truth view information, this hinders the wide application of the proposed method.
>
> - The view information, represented by the rotation matrix of the camera pose, is used during both training and inference.
>
> - In real world application, many modern systems support real-time camera calibration and pose estimation throught techniques such as Simultaneous Localization and Mapping (SLAM) [1]. By integrating these technologies, our approach ensures consistent viewpoint information, even in dynamic environments where exact camera poses may fluctuate.
>
> - Futhermore, slight inaccuracies in camera pose estimation do not significantly affect the understanding of relative object positions. In particular, inaccuracies in translation have minimal impact when rotation matrices are used to focus on the orientation. Our approach leverages rotation matrices to maintain consistency in relative object positions across different viewpoints. This ensures that VCPE remains robust to minor variations or inaccuracies in camera translations, while preserving critical spatial relationships through rotational alignment.
>
>
> >Question 2: Some metrics, such as the ROUGE-L and EM on the ScanQA dataset, are not reported.
>
> Thank you for bringing this to our attention. We have updated **Table 3** in the revised paper to include these matrics.
>
> > Question 3: Why are "near" and "far" considered viewpoint-related questions with potential ambiguities in line 519? The distance between objects will remain constant from the given viewpoint.
>
> The terms "Near" and "far" can be relative to the camera position. For example, in a description like "This chair is at the far end of the room by the fireplace," the phrase "far end" becomes inherently ambiguous, as its meaning depends on the observer's perspective. The chair referred to will vary depending on the camera's position within the room.
>
> >Question 4: Why is 'Chat-3D v2' marked with * in Tables 1, 3, and 4 but not in Table 2? Does this denote reproduced results?
>
> Thank you for pointing this out. We had missed the * in Table 2 and it has been added in the revised version of the paper.

---

> ### Comment · Reviewer_1oXE · 2024-11-25
>
> The author's responses have addressed some of my concerns; however, the proposed method still does not demonstrate a significant performance improvement compared to previous approaches. As a result, I will raise my score from 3 to 5.

---

### Official Review · Reviewer_C7XU · 2024-11-02

**Soundness:** 3
**Presentation:** 3
**Contribution:** 2
**Rating:** 5
**Confidence:** 5

**Summary:**

This paper proposes to enhance 3D LLM by incorporating information about 3D viewpoints and relative spatial distances. Besides, they create the DetailedScanRefer dataset with grounding annotations for each object described. The experimental results show the improvement over baseline methods.

**Strengths:**

1. The motivation for incorporating viewpoint and spatial distance is reasonable.
2. The DetailedScanRefer dataset with fine-grained grounding annotations is valuable for future research in 3D object grounding.
3. The experiments demonstrate the effectiveness of the proposed method.

**Weaknesses:**

1. The VCPE module's reliance on camera viewpoints as an additional input may pose challenges for real-world applications due to the difficulty of acquiring such data.
2. The spatial distance-aware loss function appears to overlook semantic similarities between objects. Objects that are spatially close but semantically different from the target should incur a greater penalty than those that are semantically similar but spatially distant. A more effective approach might involve incorporating both spatial and semantic distances into the cross-entropy loss.
3. The utility of the newly created dataset for enhancing grounding performance is questionable, as indicated by the ablation results in Table 4. Additional evidence supporting the dataset's importance would be beneficial, along with strategies to enhance its robustness, especially considering potential errors in object ID assignments by GPT-4o.

**Questions:**

Please refer to the weaknesses section.

---

> ### Author Response · Authors · 2024-11-21
> **Response to Reviewer C7XU Part 1/3**
>
> We sincerely thank the reviewer for their insightful feedback and constructive suggestions, and we will address the identified weaknesses in detail below.
>
> >The VCPE module's reliance on camera viewpoints as an additional input may pose challenges for real-world applications due to the difficulty of acquiring such data.
>
> We would like to clarify the practicality of using viewpoint information as follows:
> - **Integration with Real-Time Camera Calibration Technologies**: Modern systems support real-time camera calibration and pose estimation using techniques such as Simultaneous Localization and Mapping (SLAM) [1]. By integrating these technologies, our approach ensures consistent and reliable viewpoint information, even in dynamic real-world scenarios where exact poses may vary.
> - **Robustness to Pose Estimation Inaccuracies**: Slight inaccuracies in camera pose estimation do not significantly affect the understanding of relative positions (e.g. left-right relationships). Specifically, translation inaccuracies have minimal impact when rotation matrices are used exclusively. Our approach leverages rotation matrices to ensure that our Viewpoint-Consistent Position Encoding (VCPE) is robust to minor pose inaccuracies, preserving critical spatial relationships through rotational alignment.
>
> > The spatial distance-aware loss function appears to overlook semantic similarities between objects. Objects that are spatially close but semantically different from the target should incur a greater penalty than those that are semantically similar but spatially distant. A more effective approach might involve incorporating both spatial and semantic distances into the cross-entropy loss.
>
> The standard cross-entropy loss inherently penalizes semantic discrepancies. When the model predicts an object with *incorrect* semantics (i.e., belonging to a different category than the target), the cross-entropy loss assigns a higher penalty. This ensures that semantically incorrect predictions, whether spatially close or distant, are appropriately penalized for not matching the target's category.
>
> However, previous models like Chat3D-v2 have overlooked the explicit consideration of spatial distance during training. Our proposed Distance-Aware Cross-Entropy (DACE) loss briges this gap by incorporating spatial proximity into the loss function, enhancing the model's ability to account for both semantic and spatial alignment.

---

> ### Author Response · Authors · 2024-11-21
> **Response to Reviewer C7XU Part 2/3**
>
> >The utility of the newly created dataset for enhancing grounding performance is questionable, as indicated by the ablation results in Table 4. Additional evidence supporting the dataset's importance would be beneficial, along with strategies to enhance its robustness, especially considering potential errors in object ID assignments by GPT-4o.
>
> We would like to provide additional evidence supporting the importance of our dataset and outline the strategies we've employed to enhance its robustness, especially concerning potential errors in object ID assignments by GPT-4o.
>
> **Enhancing Grounding Performance through Effective Use of Anchor Descriptions**
>
> In the ScanRefer dataset [2], each sample description typically consists of two parts: the first sentence refers to the target object, while the subsequent sentences describe surrounding "anchor" objects. Statistical analysis indicates that including these anchor descriptions has only a minor impact on grounding accuracy, as evidenced by the results below:
>
> | Method                    | Unique (Acc\@0.25 / Acc\@0.5) | Multiple (Acc\@0.25 / Acc\@0.5) | Overall (Acc\@0.25 / Acc\@0.5) |
> |---------------------------|-----------------------------|-------------------------------|------------------------------|
> | Ours (first sentences)    | 73.52 / 46.60               | **33.71 / 21.20**                 | 41.44 / 26.12                |
> | Ours (whole descriptions) | **76.33 / 53.51**           | 32.73 / 21.11             | **41.19 / 27.40**            |
>
> These results, derived from the original ScanRefer paper [2], suggest that processing the entire description does not significantly improve performance, particularly in the "multiple" category, which accounts for the majority of the dataset (7,663 out of 9,538 samples in the validation split). This category involves scenes with multiple objects of the same type, where distinguishing between them relies heavily on contextual information from anchor objects.
>
> To address this challenge, our approach **encourages the model to effectively utilize anchor descriptions**. Instead of limiting the task to predicting only the object ID of the target, we modified it to require the model to output all object IDs mentioned in the sentence, including both target and anchor objects. This adjustment forces the model to focus on the entire description, improving its ability to disambiguate between similar objects in complex scenes. Additionally, this method enables us to generate more ground-truth data, as multiple objects are now grounded per description.

---

> ### Author Response · Authors · 2024-11-21
> **Response to Reviewer C7XU Part 3/3**
>
> **Strategies for Enhancing Dataset Robustness**
>
> To ensure the correctness of the DetailedScanRefer dataset, we have implemented several stringent data validation steps:
>
> 1. **Alignment with Ground-Truth IDs**: We ensured basic correctness by aligning the annotated objects with the ground-truth (GT) object IDs from the original ScanRefer dataset. Specifically, we removed samples where the first object ID provided by GPT-4o did not align with the target object ID. This step ensures that at least the target object is correctly recognized, enhancing the reliability of the annotations.
>
> 2. **Data Filtering**: We filtered out entries containing invalid IDs, NaN values, or IDs exceeding valid ranges. This process eliminates erroneous data points, thereby enhancing overall data quality.
>
> 3. **Annotation Reliability Assessment**: We assessed annotation reliability using the GPT-4 API's rating system, as demonstrated in Table 7 of Appendix A. This assessment provides a quantitative measure of annotation confidence and accuracy.
>
> To ensure the robustness DetailedScanRefer dataset, we have implemented the following strategy:
>
> 1. **Use Photo Instead of Rendered Image**: Instead of relying on rendered images, we utilize real-world photos, offering significantly higher quality. The comparison between rendered images and real photos could be find in Appendix 1 in our paper. When using rendered images, our data cleaning process yielded 9,659 samples. However, by switching to real photos and applying the same data cleaning process, we ended up with 16,151 data points, demonstrating the superior effectiveness and richness of using real-world photos.
>
> 2. **Careful Prompt Design**: We meticulously crafted prompts to ensure that annotations were generated only when the GPT-4o model confidently recognized objects in the descriptions. This careful design minimizes ambiguity and enhances the quality of the generated annotations.
>
> By implementing these strategies, we have significantly enhanced the robustness of the DetailedScanRefer dataset and minimized potential errors in object ID assignments by GPT-4o. Our approach not only improves the model's grounding performance by effectively leveraging anchor descriptions but also ensures the dataset's reliability for future research.
>
> We believe that these additional explanations and evidence address your concerns regarding the dataset's utility and robustness. We remain committed to further refining our methods and exploring new strategies to maximize the dataset's effectiveness in advancing grounding performance.
>
> Thank you once again for your insightful and valuable feedback.
>
>
> **Reference**
> [1] H. Durrant-Whyte and T. Bailey, "Simultaneous localization and mapping: part I," in IEEE Robotics & Automation Magazine, vol. 13, no. 2, pp. 99-110, June 2006, doi: 10.1109/MRA.2006.1638022.
> keywords: {Simultaneous localization and mapping;Mobile robots;Robotics and automation;History;Artificial intelligence;Navigation;Vehicles;Buildings;Bayesian methods;Particle filters},
>
> [2] Chen, D.Z., Chang, A.X., Nießner, M. (2020). ScanRefer: 3D Object Localization in RGB-D Scans Using Natural Language. In: Vedaldi, A., Bischof, H., Brox, T., Frahm, JM. (eds) Computer Vision – ECCV 2020. ECCV 2020. Lecture Notes in Computer Science(), vol 12365. Springer, Cham. https://doi.org/10.1007/978-3-030-58565-5_13

---

### Official Review · Reviewer_wme5 · 2024-11-02

**Soundness:** 3
**Presentation:** 2
**Contribution:** 3
**Rating:** 5
**Confidence:** 3

**Summary:**

This work begins by emphasizing the importance of viewpoint in grounding and introduces a geometry-enhanced vision-language model. It includes the VCPE and DACE modules, which address ambiguity caused by varying viewpoints and the positional information neglect in cross-entropy loss, respectively. Additionally, this work utilizes GPT-4 to generate the DetailedScanRefer dataset to support training, enabling the model to achieve substantial improvements over the baseline.

**Strengths:**

1. The motivation of this paper is clear, enhancing LLM-based 3D visual grounding models from the perspectives of viewpoint and relative spatial relationships, which aligns well with human intuition.
2. This paper introduces the integration of viewpoint information into the LLM-based paradigm and designs a loss function that better aligns with the task. Additionally, it presents a fine-grained dataset, rich in content, which makes a commendable contribution to the community.
3. The method proposed in this paper achieves significant improvements over the baseline.

**Weaknesses:**

1. The experimental section lacks a sufficiently comprehensive comparison. In addition to the baseline and other LLM-based models, comparisons should also be made with models specifically designed for this task to better demonstrate the overall performance of the proposed model in this context.
2. Since this method uses viewpoint information as model input, it is important to clarify whether other comparison models also include or utilize this information to ensure fairness. This is particularly relevant for works like ConcreteNet (“Four Ways to Improve Verbo-visual Fusion for Dense 3D Visual Grounding,” ECCV 2024), which already explores viewpoint information. A comparison with this model's use of viewpoint information would be valuable.
3. From the ablation study shown in Table 4, it is observed that most of the performance gains are concentrated in the DACE module. It remains unclear whether similar improvements could be achieved using only "world + DetailedScanRefer + DACE." The necessity of incorporating viewpoint information is not well substantiated. Additionally, it appears that DetailedScanRefer does not contribute to performance improvement; clarification on this would be helpful. I would also like to see whether DACE remains effective without DetailedScanRefer.
4. There is no quantitative metric has been provided to assess the quality of the dataset generated by GPT-4 and Mask3D.

**Questions:**

See weaknesses.

---

> ### Author Response · Authors · 2024-11-21
> **Response to Reviewer wme5 Part 1/4**
>
> We sincerely thank the reviewer for their insightful feedback and constructive suggestions, and we will address the identified weaknesses in detail below.
>
> >The experimental section lacks a sufficiently comprehensive comparison. In addition to the baseline and other LLM-based models, comparisons should also be made with models specifically designed for this task to better demonstrate the overall performance of the proposed model in this context.
>
> We have included a comprehensive performance comparison table in **Appendix C**, which summarizes the performance of various models on the ScanRefer benchmark since 2022. The table now cover expert models, general backbones and other unified 3D LLMs, offering a detailed comparison with our proposed model, GeVLM.
>
> In the main paper, we have updated **Table 1** to include comparisons with 3DJCG and D3Net. These models were selected to ensure a fair comparison among unified 3D-Language Models (3D-LLMs) capable of handling multiple tasks.
>
> ### Table: Performance comparison of other models on ScanRefer.
>
> | Method                    | Unique (Acc\@0.25 / Acc\@0.5) | Multiple (Acc\@0.25 / Acc\@0.5) | Overall (Acc\@0.25 / Acc\@0.5) |
> |---------------------------|-----------------------------|-------------------------------|------------------------------|
> | **ScanRefer**                 | 76.33 / 53.51               | 32.73 / 21.11                 | 41.19 / 27.40                |
> | **MVT**                       | 77.67 / 66.45               | 31.92 / 23.30                 | 39.43 / 33.26                |
> | **3D-SPS**                    | 84.12 / 66.72               | 40.32 / 29.82                 | 48.82 / 36.98                |
> | **ViL3DRel**                  | 81.58 / 68.62               | 40.30 / 30.71                 | 47.94 / 37.73                |
> | **BUTD-DETR**                 | 84.20 / 66.30               | 46.60 / 35.10                 | 52.20 / 39.80                |
> | **HAM**                       | 79.24 / 67.86               | 41.46 / 34.03                 | 48.79 / 40.60                |
> | **3DRP-Net**                  | 83.13 / 67.74               | 42.14 / 31.95                 | 50.10 / 38.90                |
> | **EDA**                       | 85.76 / 68.57               | 49.13* / 37.64             | 54.59 / 42.26                |
> | **M3DRef-CLIP**               | 85.30 / 77.20               | 43.80 / 36.80                 | 51.90 / 44.70                |
> | **ConcreteNet**               | 86.40* / 82.05*           | 42.41 / 38.39                 | 50.61 / 46.53*            |
> | **DORa**                      | - / -                       | - / -                         | 52.80* / 44.80            |
> | **3D-VisTA** (need task fine tuning)            | 81.60 / 75.10               | 43.70 / 39.10*             | 50.60 / 45.80                |
> | **3D-VLP**(need task fine tuning)                    | 84.23 / 64.61               | 43.51 / 33.41                 | 51.41 / 39.46                |
> | 3DJCG      (grounding + captioning)               | - / 64.34               | -/ 30.82                 | - / 37.33                |
> | D3Net      (grounding + captioning)                 | - / 72.04                   | - / 30.05                     | - / 37.87                    |
> | GeVLM (Ours)              | 82.00 / 75.70               | 39.00 / 34.70                 | 46.90 / 42.30                |
>
> Methods highlighted in **bold** are expert models designed specifically for the 3D grounding task, limiting their capabilities to this single task. In contrast, models like 3D-VLP and 3D-VisTA serve as general backbones for a variety of 3D tasks but require task-specific fine-tuning to perform competitively on individual tasks.
>
> - Key observations from the table:
>   - **Competitive Accuracy**: GeVLM surpasses all unified 3D-LLM and demonstrates strong performance among expert models, particularly in the  Acc\@0.5 metric. This highlights the effectiveness of our DACE method in aligning predicted and reference locations with minimized distance-based penalties.
>   - **No Task-Specific Fine-Tuning**: Unlike the expert models listed, GeVLM achieves its results without requiring task-specific fine-tuning. This underscores the robustness and generalization capabilities of our model compared to specialized models that are heavily optimized for specific tasks.
>   - **Broad Applicability**: While all listed models are either expert systems tailored to a specific task or require task specific fine-tuning, GeVLM is versatile, addressing a diverse range of tasks. This flexibility emphasizes the broader utility of our approach, distinguishing it from single-purpose models, and making its results even more significant.

---

> > ### Author Response · Authors · 2024-11-21
> > **Response to Reviewer wme5 Part 2/4**
> >
> > >Since this method uses viewpoint information as model input, it is important to clarify whether other comparison models also include or utilize this information to ensure fairness. This is particularly relevant for works like ConcreteNet (“Four Ways to Improve Verbo-visual Fusion for Dense 3D Visual Grounding,” ECCV 2024), which already explores viewpoint information. A comparison with this model's use of viewpoint information would be valuable.
> >
> > - **Novel Use of Viewpoint Information**: To the best of our knowledge, current unified 3D LLMs capable of performing grounding tasks, such as 3D-LLM and Chat-3D v2, do not incorporate viewpoint-related information, specifically camera poses. Our approach introduces a novel aspect by leveraging rotated coordinates to effectively address viewpoint-dependent tasks, particularly in 3D grounding task.
> > - **Comparison with ConcreteNet**: We appreciate the mention of ConcreteNet, which incorporates viewpoint information using a different approach - a Global Camera Token that encodes camera-related features, supervised by actual camera positions. Interestingly, an ablation study in ConcreteNet showed that incorporating rotation information into this token resulted in a performance drop, which is counterintuitive given the expectation that rotation awareness would enhance understanding of viewpoint-dependent tasks.
> > - **Efficient Inference Through Rotated Coordinates**: ConcreteNet employs multiple rotated views of the scene to make predictions, refining mask outputs based on aggregated predictions from these views. In contrast, our approach directly utilizes rotated coordinates, bypassing  the need for translation vectors or ensembling strategies. This results in a more computationally efficient inference process.
> >
> > - **Key Differences and Our Approach's Rationale**: Unlike ConcreteNet, which emphasizes *translation* vectors in its viewpoint-related features, our approach focuses on *rotation* matrices. As shown in Table 5 of the paper, incorporating rotation matrices resulted in improved performance compared to using camera coordinates.  We attribute this improvement to the alignment provided by rotation matrices, as they effectively account for orientation changes in a way that translation vectors alone cannot. While *translation* adjusts only the scene’s position along the axes, *rotation* fundamentally transforms how view-dependent descriptions are interpreted, leading to a more robust understanding of spatial relationships.

---

> ### Author Response · Authors · 2024-11-21
> **Response to Reviewer wme5 Part 3/4**
>
> >From the ablation study shown in Table 4, it is observed that most of the performance gains are concentrated in the DACE module. It remains unclear whether similar improvements could be achieved using only "world + DetailedScanRefer + DACE." The necessity of incorporating viewpoint information is not well substantiated.
>
> We conducted experiments comparing the "World + DetailedScanRefer + DACE" configuration to our proposed model, GeVLM incorporates rotated scene coordinates (utilizing viewpoint information), DetailedScanRefer, and DACE. The results across various tasks and metrics are summarized as follows:
>
> | Metric                                | World + DetailedScanRefer + DACE | GeVLM                |
> |--------------------------------------|----------------------------------|----------------------|
> | **ScanRefer (Acc\@0.25)**             | **47.28**                            | 46.94                |
> | **ScanRefer (Acc\@0.50)**             | **42.35**                            | 42.29                |
> | **ScanRefer (Unique Acc\@0.25)**      | 81.41                            | **82.04**                |
> | **ScanRefer (Unique Acc\@0.50)**      | 75.33                            | **75.72**                |
> | **ScanRefer (Multiple Acc\@0.25)**    | **39.53**                            | 38.97                |
> | **ScanRefer (Multiple Acc\@0.50)**    | **34.87**                            | 34.70                |
> | **Multi3dRefer (F1\@0.25)**           | 49.18                            | **49.95**                |
> | **Multi3dRefer (F1\@0.50)**           | 45.33                            | **46.14**                |
> | **ScanQA (CIDEr)**                   | 87.73                            | **90.53**                |
> | **ScanQA (Bleu-4)**                  | 13.79                            | **15.45**                |
> | **SQA3D (EM)**                       | **52.53**                            | 52.24                |
>
> While the "World + DetailedScanRefer + DACE" configuration shows slightly better performance on ScanRefer Multiple and Overall metrics(Acc\@0.25 and Acc\@0.50), our GeVLM model excels in the Unique category, achieving higher Unique Acc\@0.25 and Unique Acc\@0.50. Additionaly, GeVLM outperforms in tasks such as Multi3dRefer and ScanQA.
>
> The use of rotated scene coordinates aligns the spatial representation with the camera's viewpoint. This alignment proves advantageous, especially for tasks like ScanQA and SQA3D, which do not provide explicit camera poses and typically rely on world coordinates.  We have clarified this point in the experimental section.  By rotating the coordinates, we standardize the spatial representation across tasks, enabling the model to generalize more effectively and leverage learned spatial relationships.
>
> Considering the overall performance across multiple tasks, we find that incorporating viewpoint information through the rotated coordinate system offers significant benefits. Thus, we continue to employ rotated coordinates in our GeVLM model to enhance its generalization capabilities and performance across diverse 3D language tasks.

---

> ### Author Response · Authors · 2024-11-21
> **Response to Reviewer wme5 Part 4/4**
>
> >Additionally, it appears that DetailedScanRefer does not contribute to performance improvement; clarification on this would be helpful.
>
> DetailedScanRefer was introduced to address the limited annotation of anchor objects in ScanRefer by providing explicit annotations that highlight these objects. This encourages the model to process spatial context rather than solely focusing on target identification. These additional details enrich the model’s comprehension of scene relationships and improve alignment with referring expressions.
>
> As shown in Table 4 (comparing row 5 to row 6), the impact of DetailedScanRefer is most pronounced when used in combination with DACE. DACE reinforces spatial awareness by penalizing the model for failing to accurately recognize both the target and anchor objects.
>
> This highlights that the combination of DetailedScanRefer and DACE significantly enhances the model’s effectiveness by promoting more precise spatial reasoning and anchor-object recognition, which are critical for tasks involving complex referring expressions.
>
>
> >I would also like to see whether DACE remains effective without DetailedScanRefer.
>
> We have added an ablation for **VCPE + DACE** in the table below in for clarity:
>
> | VCPE    | Detailed | DACE      | Unique (Acc\@0.25) | Unique (Acc\@0.50) | Multiple (Acc\@0.25) | Multiple (Acc\@0.50) | Overall (Acc\@0.25) | Overall (Acc\@0.50) |
> |---------|----------|-----------|-------------------|-------------------|---------------------|---------------------|-------------------|-------------------|
> | Rotate  | --       | --        | 79.6              | 74.7              | 36.2                | 32.6                | 44.2              | 40.4              |
> | Rotate  | --       | $T$=0.05  | 79.5              | 73.7              | 37.9                | 33.7                | 45.6              | 41.1              |
>
> As shown in the table, incorporating DACE without DetailedScanRefer leads to a noticeable performance improvement across several metrics, including increases in **Multiple Acc\@0.25**, **Multiple Acc\@0.50**, and overall metrics. This demonstrats the effectiveness of DACE, even when DetailedScanRefer is not utilized.
>
>
> >There is no quantitative metric has been provided to assess the quality of the dataset generated by GPT-4 and Mask3D.
>
> We have provided the quantitative metric regarding the quality of the Dataset in appendix **A.4 DATASET QUALITY EVALUATION**.
>
> Additionally, we have added a table summarizing key statistics and metrics for Mask3D's performance across both the training and validation splits. This includes counts for IoU ≥ 0.25 and IoU ≥ 0.50, as well as maximum IoU rates, to further illustrate the quality and comprehensiveness of the generated proposals.
>
> | Metric                                  | Train Split Count | Validation Split Count |
> |-----------------------------------------|-------------------|-----------------------|
> | **Total Count (Original ScanRefer Dataset)** | 36,665            | 9,508                 |
> | **IoU ≥ 0.25 Count**                    | 36,187            | 8,924                 |
> | **IoU ≥ 0.50 Count**                    | 35,061            | 8,168                 |
> | **Max IoU\@0.25**                        | 98.70%            | 93.86%                |
> | **Max IoU\@0.50**                        | 95.63%            | 85.91%                |
>
> During the trainig process, we utilize only 32,338 annotations that meet the strict criterion of IoU≥0.75 with ground truth objects. This high threshold ensures that only highly accurate object proposals are retained, emphasizing the precision and relevance of the dataset for effective downstream tasks.

---

> ### Comment · Reviewer_wme5 · 2024-12-03
>
> Thanks for the detailed responses and the authors' efforts. However, based on the experimental results, it appears that GeVLM still exhibits a significant performance gap compared to the current state-of-the-art (SOTA), particularly in terms of Acc@0.25, where the disparity is especially pronounced. As a paper explicitly titled “3D Object Grounding”, such a performance gap is difficult for me to accept.
>
> Additionally, the experiments with “World + DetailedScanRefer + DACE” further raise questions about the contributions of this work. In the ScanRefer dataset, “Multiple” category data accounts for approximately 80%, and GeVLM clearly struggles to handle this critical aspect effectively. This limitation also explains why GeVLM underperforms in the Overall metric in ScanRefer.
>
> In conclusion, my core concern has not been resolved, and I will maintain my original score of 5.

---

### Note · Authors · 2024-12-15

I have read and agree with the venue's withdrawal policy on behalf of myself and my co-authors.